# Evaluation of Cognitive Functions in People Living with HIV Before and After COVID-19 Infection

**DOI:** 10.3390/v17010135

**Published:** 2025-01-20

**Authors:** Dimtrios Basoulis, Elpida Mastrogianni, Irene Eliadi, Nikolaos Platakis, Dimitris Platis, Mina Psichogiou

**Affiliations:** 1st Department of Internal Medicine, Laiko General Hospital, 11527 Athens, Greece; elpidamastrogianni@gmail.com (E.M.); eirini.iliadi@gmail.com (I.E.); nickplat7@gmail.com (N.P.); platisdim27@gmail.com (D.P.)

**Keywords:** HIV, COVID-19, neurocognitive function, aging

## Abstract

Background: Cognitive function decline is a problem in aging people living with HIV (PLWHIV). COVID-19 infection is associated with neuropsychiatric manifestations that may persist. The aim of our study was to evaluate cognitive function in PLWHIV before and after COVID-19 infection. Methods: This was a prospective observational study conducted at “Laiko” General Hospital from July 2019 to July 2024. The Montreal Cognitive Assessment (MOCA) scale was used to evaluate cognitive functions. Results: 116 virally suppressed PLWHIV participated (mean age: 47.6 years, 91.4% male); 60 underwent repeated evaluation after the pandemic at a median interval of 3.1 years. The median MOCA score was 24 (22–26), with 35.3% scoring within normal limits. A negative correlation was observed between MOCA scores and age (ρ = −0.283, *p* = 0.002), but not with a CD4 count at diagnosis (ρ = 0.169, *p* = 0.071) or initial HIV RNA load (ρ = 0.02, *p* = 0.984). In the subgroup with repeated testing, MOCA was correlated with the CD4 count (ρ = 0.238, *p* = 0.069 in the first and ρ = 0.319, *p* = 0.014 second test). An improvement in performance was observed (median score increase from 24 to 25, *p* = 0.02). Conclusions: MOCA can detect early changes in cognitive function in PLWHIV. Further studies are required to determine the role of COVID-19 over time.

## 1. Introduction

The development and widespread use of novel antiretroviral medicines has changed the prognosis of HIV [1]. The effects of HIV infection on neurocognitive functions have remained a topic of concern in spite of the advances in treatment [2]. HIV has been shown to bypass the blood–brain barrier initiate and maintain neuroinflammation and survive in latent form within the central nervous system establishing a viral depot. Moreover, as people living with HIV (PLWHIV) continue to age, other causes of cognitive impairment related to the aging process are added as well as added medication toxicity.

Recently, a consensus statement was published attempting to differentiate between different states of brain injury in relation to HIV, medications and other conditions [3]. The authors suggest that HIV-associated brain injury (HABI) should be considered as one of different causes of cognitive impairment in PLWHIV and that it should be further differentiated between legacy or active brain injury on the basis of HIV RNA load detection. The authors furthermore make a distinction between low scores on various cognition tests and the clinical signs of cognitive impairment. Nonetheless, tools designed for screening for cognitive impairment can provide early warning signs and prompt further investigation [4].

The British HIV Association recommends that cognitive dysfunction screening should be available for all individuals within three months of an HIV diagnosis [5]. Additionally, all PLWHIV should undergo regular screening after events that might trigger or worsen cognitive impairment; otherwise, annual screening is advised. Similarly, the Infectious Diseases Society of America offers comparable guidelines [6]. The World Health Organization supports the routine screening and management of mental health disorders for key populations of PLWHIV to improve the health outcomes and adherence to combined antiretroviral treatment (cART), but it does not provide details with regard to the methods used for this [7].

In recent years, another virus has dominated scientific interest with the emergence of SARS-CoV-2 and the ensuing pandemic. In fact, it has been shown that HIV infection has been associated with increased frequency and severity in SARS-CoV-2 infections [8]. COVID-19 has been linked to acute and long-term neurocognitive decline [9,10]. On the other hand, there have been studies comparing neurocognitive symptoms and biomarkers in COVID-19 patients in recovery and healthy subjects and found no significant differences [11]. In a recently published meta-analysis, the authors point out that even though neurocognitive screening tests are scored lower for persons with post-acute COVID-19 symptomatology, they failed to attribute any clinical significance to these findings [12].

The Montreal Cognitive Assessment (MOCA) scale is an easy-to-use very sensitive screening tool for the diagnosis of cognitive impairment in several neurological and non-neurological conditions [4,13,14,15]. In HIV research, it has been shown that compared to non-HIV persons, PLWHIV have systematically scored lower when tested, prompting to question the mild cognitive impairment reference range [16]. Nonetheless, as a tool to monitor an individual’s progress, it remains useful. In COVID-19, MOCA testing has been the mainstay of neurocognitive assessment [17].

This study was initially designed to assess the MOCA scores longitudinally in a population of virally suppressed PLWHIV and provide associations with viral characteristics, opportunistic infections and treatments. The emergence of the pandemic shifted our focus and provided a unique opportunity to compare data collected before and after COVID-19 and examine whether SARS-CoV-2 infection might contribute to cognitive deterioration in this population.

## 2. Materials and Methods

### 2.1. Study Design

This was a prospective study of people living with HIV followed at the First Dept of Internal Medicine, Laiko General Hospital, National and Kapodistrian University of Athens. The study period spanned from 4 July 2019 to 4 July 2024. The primary outcome was defined as changes in the MOCA score in relation to HIV history. Secondary outcomes included investigating MOCA scores at different time points and whether previous SARS-CoV-2 infection affected the results.

Patients were randomly selected from within a cohort of individuals with available cryopreserved peripheral blood mononuclear cell (PBMC) samples available before cART and who had been followed for at least two years after cART initiation and had achieved a sustained virological response as described in previous work [18]. Participants available for follow-up testing after the pandemic were used for comparison analysis before and after possible SARS-CoV-2 infection.

A sustained virological response was defined as a viral load lower than 50 copies/mL in two consecutive measurements after cART initiation retained throughout the follow-up. The cognitive function was measured using the MOCA scale [15]. This test has been previously validated in Greek populations with dementia and Parkinson’s disease [19,20]. This test has also been validated in an HIV cohort, and while it seemed to be under-performing, the authors suggested the use of the same cut-off points [21]. The authors received online training and obtained permission for research use. The participants were screened during their follow-up testing for anxiety or depression using the Hospital Anxiety and Depression scale (HADS) [22].

A COVID-19 diagnosis was established through the existence of a history of positive PCR tests as evidenced in the Greek National COVID-19 registry. The severity of infection was scored using the WHO published scale [23]. COVID-19 symptoms were self-reported by the participants.

Other covariates included age, gender, educational level, transmission group, AIDS status, pre-cART HIV RNA levels in plasma and DNA levels in PBMCs, CD4 cell counts at diagnosis and at each testing date and the cART regimen, and a history of neurological or psychiatric diagnoses.

All the participants were included after providing written informed consent. This study has been approved by the Ethical Review Board of Laiko General Hospital, Athens, Greece. This article is a revised and expanded version of a paper entitled “Evaluation of cognitive functions in people living with HIV before and after COVID-19”, which was presented at the 36th Panhellenic AIDS Conference, 29 November–1 December 2024, Athens, Greece [24].

### 2.2. Statistical Analysis

Categorical variables are expressed by absolute values and relative frequencies. Continuous variables were measured via mean and standard deviation (SD), for normally distributed variables, or the median and interquartile range (IQR), for non-normally distributed variables. Normality was tested using the Shapiro–Wilks test and the equality of variances by using Levene’s test. Group comparisons for unrelated continuous variables were performed using the Mann–Whitney U test for continuous non-normally distributed variables and Student’s t-test for normally distributed variables. For related continuous normally distributed variables, comparisons were performed using the paired Student’s *t*-test and for non-normally distributed variables by using Wilcoxon’s test. Categorical related variable comparisons were examined using McNemar’s test. The associations between continuous variables were examined using Pearson (r) correlation for normally distributed variables and Spearman (ρ) for non-normally distributed variables. Group comparisons for categorical variables were performed using the Chi-square and ANOVA tests. Group comparisons at different timepoints were tested using a mixed ANOVA method. The level of statistical significance was set to <0.05. Statistical analysis was performed using the IBM SPSS Statistics for Windows, version 25.0 (2017, IBM Corp., Armonk, NY, USA).

## 3. Results

One hundred and sixteen virally suppressed PLWHIV had at least one MOCA test available for evaluation. The main demographic and clinical characteristics are presented in Table 1.

The participants were overwhelmingly male (91.4%) with a mean age of 47.6 years at the time of taking the test. The median test result was 24 with only 41/116 (35.3%) participants scoring within the normal range and the majority 67/116 (57.8%) scoring within the mild cognitive impairment bracket. There was a significant inverse relation between age and MOCA score (ρ = −0.283, *p* = 0.002). We also identified a significant correlation between the final score and CD4 counts at the time of testing (ρ = 0.296, *p* = 0.002), while we did not identify any correlation with the MOCA score and CD4 counts at HIV diagnosis (ρ = 0.169, *p* = 0.071), initial HIV RNA (ρ = 0.02, *p* = 0.984) or DNA (ρ = −0.059, *p* = 0.674) loads, years with HIV (ρ = −0.109, *p* = 0.248) or AIDS diagnoses (ρ = 0.1, *p* = 0.614). The MOCA score < 25 was not associated with gender (*p* = 1), history of psychiatric (*p* = 1) or neurological conditions (*p* = 0.159), type of backbone treatment (*p* = 0.886), type of base treatment (*p* = 0.945) or transmission mode (*p* = 0.238).

Sixty PLWHIV returned for a follow-up test after the pandemic, after a median of 3.1 (2.7–3.5) years. Demographic and clinical characteristics of this subpopulation are presented in Table 2. In this sub-cohort, we did not identify correlations between scores at either timepoint and age (for the two testing timepoints, respectively, ρ = −0.150, *p* = 0.253; ρ = −0.099, *p* = 0.451), years with HIV (ρ = −0.093, *p* = 0.481; ρ = −0.037, *p* = 0.782) or years with AIDS (ρ = 0.406, *p* = 0.119; ρ = 0.193, *p* = 0.474). There was an association trend between CD4 counts and the MOCA score at the first testing timepoint (ρ = 0.238, *p* = 0.069) which became statistically significant as time progressed at the second timepoint (ρ = 0.319, *p* = 0.014). There was also no correlation between the MOCA scale score at the 2nd testing and neither depression (ρ = 0.07, *p* = 0.594) nor anxiety (ρ = −0.15, *p* = 0.251) as measured by using the HADS scale. There was a significant correlation (ρ = 0.516, *p* < 0.001) between anxiety and depression scores, however. Finally, there was a significant correlation (ρ = 0.465, *p* < 0.001) between the MOCA scores as measured at the two timepoints.

We compared the MOCA test results between the two timepoints (Table 3). We identified a slight improvement in the scores at the second testing (median 24 vs. 25, *p* = 0.02). Investigating the various sections of the MOCA score, there was a slight improvement in recollection, especially when scored using the memory index (*p* = 0.002), and there were small overall improvements, albeit not reaching statistical significance in naming (*p* = 0.083) and orientation (*p* = 0.083). The changes in the MOCA score for each participant are depicted in Figure 1.

We further compared the groups of participants with two tests who had a history of COVID-19 (n = 37) and those who did not (n = 23) in an attempt to investigate if COVID-19 contributed negatively in the MOCA scoring. The comparison data are presented in Table 4. The median time in months from COVID-19 diagnosis to testing was 9 [4,5,6,7,8,9,10,11,12,13,14,15,16,17,18,19] months. At the time of testing, no participant reported long COVID symptoms. We did not identify any statistically significant differences between the COVID-19 and the non-COVID-19 groups with regard to HIV history, HADS and MOCA scores. We also performed a mixed ANOVA analysis to further investigate the potential effect of SARS-CoV-2 infection on the changes in MOCA scores through time. There was marginally no statistically significant difference in scoring over time (F = 3.89, *p* = 0.053), nor differences in scoring dependent on previous COVID-19 (F = 0.049, *p* = 0.826), nor differences in scoring dependent on the interaction between a history of COVID-19 and the testing timepoint (F = 2.5, *p* = 0.119). Figure 2 depicts the estimated marginal means of MOCA scores in relation to COVID-19 infection over the two timepoints of testing. Adding age at the second test to the model did not significantly alter the results (Appendix A).

## 4. Discussion

In our study, we attempted to assess the cognitive functions in PLWHIV both with regard to time progression and with potential exposure to SARS-CoV-2. We were fortunate in the sense that we had begun a longitudinal study of MOCA scores in our population before the pandemic started, and we were in a position afterwards to compare persons with and without a history of COVID-19 and evaluate its impact. Our results show that COVID-19 did not seem to have a negative effect on the cognition in our limited population.

A large proportion of the participants (57.8%) was identified as having mild cognitive impairment. This finding is in agreement with established literature associating HIV infection with a faster neurocognitive decline compared with uninfected persons as measured using a variety of methods including neurocognitive scores and MRI-based scores [25,26]. Of note, none of the participants in our cohort with mild impairment reported symptoms or limitations in their daily life. Viral failure has been associated with both symptomatic and non-symptomatic neurological decline [27]. The correlation with CD4 counts is more nuanced with several studies showing conflicting results. In our cohort, we identified a correlation of current CD4 counts with MOCA scores. In previous works, it appears that such a correlation has been at times identified but not sustained throughout a longitudinal observation cohort from the US [28]. Moreover, the authors of this study did not identify a correlation with nadir CD4 counts; similarly, we did not identify a correlation with starting CD4 counts [28]. In a Spanish cohort, the authors attempted to create a diagnostic algorithm to screen for cognitive impairment based on nadir CD4 counts and current CD4 counts in experienced patients [29]. Others have identified an association between neurocognitive decline and a decrease in the CD4/CD8 ratio [30]. Finally, in our study, we also examined if there exists an association between scores and HIV DNA levels without identifying such. A relationship between the two parameters has been described among treatment naïve individuals [31].

In our cohort, we did not identify a cognitive decline amongst COVID-19 survivors compared to persons without a COVID-19 history. A large, published cohort attempted to compare the cognitive decline before and after SARS-CoV-2 infection in PLWHIV and uninfected persons. In this study, PLWHIV had a higher prevalence of dementia and cognitive concerns and in a longitudinal follow-up had an increased mortality risk [32]. Two studies from California have identified an increased risk of long COVID sequalae in PLWHIV with specifically a 2.5-fold greater risk of neurocognitive symptoms [33,34]. To our knowledge, a study examining neurocognitive abilities longitudinally before and after COVID-19 among PLWHIV has not been published so far. It is likely that our sample size was too small to identify potential effects, in spite of the presumed high prevalence of long COVID symptoms. It is also possible that such symptoms would be more prominent in patients with serious SARS-CoV-2 infections.

Our study of course faces limitations. First of all, the sample size was relatively small and came from a single center only, potentially limiting the generalizability of our findings. MOCA is a sensitive and easy-to-use tool for neurocognitive decline screening, but it does not offer in-depth exploration of which domains are deficient. This would warrant more extensive tests performed by specialists, which was beyond the scope of our study. Not all the participants returned for testing in spite of our efforts. Our plan is to continue longitudinal neurocognitive evaluation in our cohort to assess for changes in the future and strengthen our findings. We also have no means to safely account for potential asymptomatic COVID-19. Patients were periodically tested during their visits to our department, but it is possible that an asymptomatic infection could have escaped detection. In theory, serological testing could be used to identify asymptomatic cases, but we would still be unable to know when in the course of the study they were infected. Nonetheless, it should be noted that of symptomatic COVID-19 patients, only 2.2% experienced chronic cognitive symptoms 3 months after infection, according to a meta-analysis of 1.2 million cases [35].

## 5. Conclusions

The MOCA scale can be effective in detecting early changes in the cognitive function in PLWHIV. Neurocognitive decline is associated with age and potentially CD4 counts at the time of testing but not with starting CD4 counts and HIV RNA or DNA levels. Further studies are required to determine the role of previous COVID-19 and other factors over time.

## Figures and Tables

**Figure 1 viruses-17-00135-f001:**
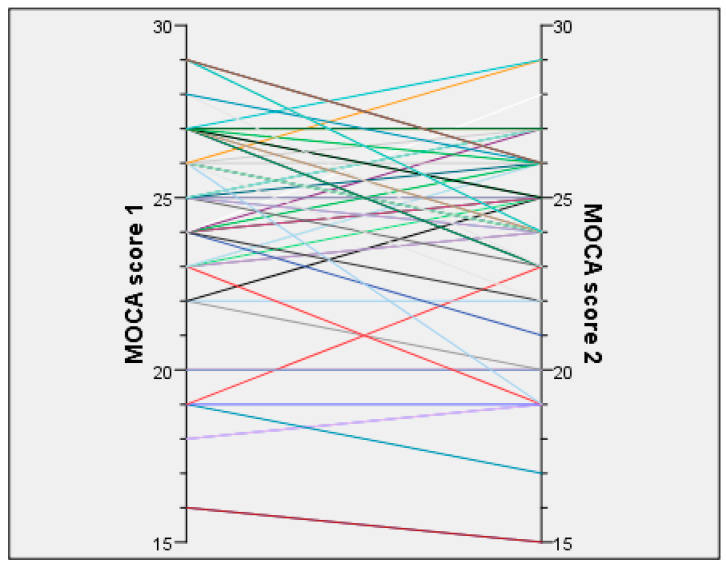
Individual change plot depicting changes in MOCA scores between the two timepoints for each participant.

**Figure 2 viruses-17-00135-f002:**
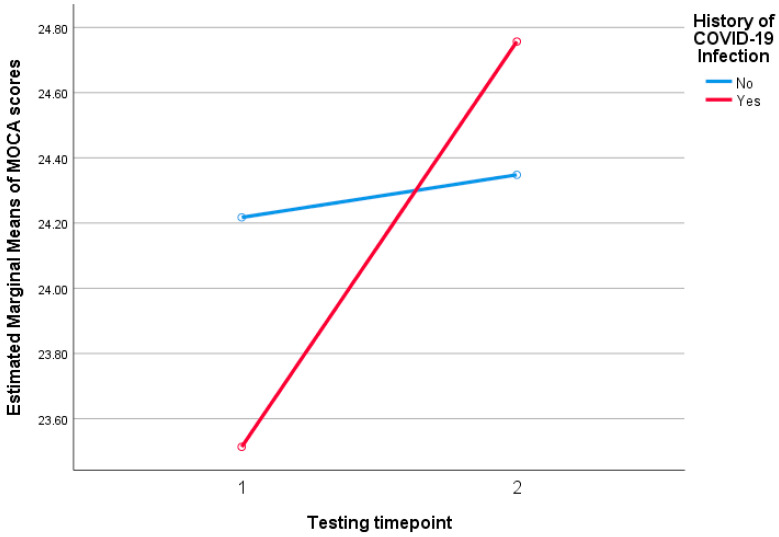
Estimated marginal means of MOCA scores over two timepoints for individuals with and without a history of COVID-19. No significant differences were observed over time, between groups or in the interaction between time and COVID-19 history (all *p* > 0.05). MOCA: Montreal Cognitive Assessment, COVID-19: coronavirus disease 2019.

**Table 1 viruses-17-00135-t001:** Baseline demographic and clinical characteristics of participants with at least one available MOCA test.

	N = 116
Male gender	106 (91.4)
Age at testing	47.6 ± 10.9
Education < 13 years	38 (32.8)
HIV acquisition	
MSM	83 (71.6)
PWID	11 (9.5)
Heterosexual	14 (12.1)
Unknown	6 (5.2)
Other	2 (1.7)
Psychiatric history	18 (15.5)
Neurological history	5 (4.3)
CD4 count at HIV diagnosis	356 (176–520)
Log HIV RNA at HIV diagnosis	4.8 ± 0.9
Log HIV DNA at HIV diagnosis	2.6 ± 0.6
AIDS diagnosis concurrent with HIV	17 (14.7)
CD4 count at testing	814 (560–984)
AIDS diagnosis at testing	29 (25)
Years with HIV at testing	8 (6–12)
Years with AIDS at testing	6.5 (3–9)
HIV Treatment	
Backbone	
ABC/3TC	11 (9.5)
TXF/XTC	100 (86.2)
Other	5 (4.3)
3rd agent	
INSTI	62 (53.4)
Boosted PI	10 (8.6)
NNRTI	39 (33.6)
Other/combination	3 (2.6)
MOCA test	
Visuospatial	4 (3–5)
Naming	3 (3–3)
Attention	6 (5–6)
Language	2 (1–3)
Abstraction	1 (0–2)
Memory Index Score	11 (8–13)
Orientation	6 (6–6)
Final Score	24 (22–26)
Normal cognitive performance (>25)	41 (35.3)
Mild impairment (18–25)	67 (57.8)
Moderate impairment (10–17)	8 (6.9)
Severe impairment (<10)	0 (0)

Values are presented as n (%) for categorical variables, mean ± standard deviation for normally distributed and median (interquartile range 25th–75th) for non-normally distributed continuous variables. MSM: men who have sex with men, PWID: people who inject drugs, ABC: abacavir, 3TC: lamivudine, TXF: tenofovir alafenamide or disoproxil, FTC: emtricitabine, INSTI: integrase strand transfer inhibitor, PI: protease inhibitor, NNRTI: non-nucleoside reverse transcriptase inhibitor, MOCA: Montreal Cognitive Assessment.

**Table 2 viruses-17-00135-t002:** Baseline demographic and clinical characteristics of participants with two available MOCA tests.

	N = 60
Male gender	55 (91.7)
Education < 13 years	19 (31.7)
HIV acquisition	
MSM	48 (80)
PWID	3 (5)
Heterosexual	7 (11.7)
Uknown	1 (1.7)
Other	1 (1.7)
Psychiatric history	11 (18.3)
Neurological history	2 (3.3)
CD4 count at HIV diagnosis	347 (138–553)
Log HIV RNA at HIV diagnosis	4.8 ± 0.9
Log HIV DNA at HIV diagnosis	2.7 ± 0.6
AIDS diagnosis concurrent with HIV	9 (15)
HIV Treatment	
Backbone	
ABC/3TC	6 (10)
TXF/XTC	52 (86.7)
Other	2 (3.3)
3rd agent	
INSTI	36 (60)
Boosted PI	5 (8.3)
NNRTI	16 (26.7)
Other/combination	1 (1.7)
COVID-19 history	37 (61.7)
Anosmia	16 (26.7)
Brain Fog	3 (5)
Severity	
Asymptomatic	4 (6.7)
Symptomatic, no treatment	28 (46.7)
Symptomatic, treated	3 (5)
Hospitalized	2 (3.3)
HADS scale	
Depression, median score	4 (1–5)
0–7	53 (88.3)
8–10	5 (8.3)
11–15	2 (3.3)
≥16	0 (0)
Anxiety, median score	4 (2–7)
0–7	49 (81.7)
8–10	5 (8.3)
11–15	5 (8.3)
≥16	1 (1.7)

Values are presented as n (%) for categorical variables, mean ± standard deviation for normally distributed and median (interquartile range 25th–75th) for non-normally distributed continuous variables. MSM: men who have sex with men, PWID: people who inject drugs, ABC: abacavir, 3TC: lamivudine, TXF: tenofovir alafenamide or disoproxil, FTC: emtricitabine, INSTI: integrase strand transfer inhibitor, PI: protease inhibitor, NNRTI: non-nucleoside reverse transcriptase inhibitor, COVID-19: coronavirus disease 2019, HADS: hospital anxiety and depression scale, MOCA: Montreal Cognitive Assessment.

**Table 3 viruses-17-00135-t003:** Comparison between the two MOCA tests.

	MOCA Test 1 (n = 60)	MOCA Test 2 (n = 60)	*p*
Age at testing	46.8 ± 10.2	49.9 ± 10.3	**<0.001**
CD4 at testing	817 (578–984)	812 (566–978)	0.301
AIDS diagnosis at testing	17 (28.3)	17 (28.3)	1
Years with HIV	8 (5–12)	11 (8–15)	**<0.001**
Years with AIDS	4 (2–9)	8 (5–12)	**<0.001**
Years under treatment	7 (4–10)	10 (8–14)	**<0.001**
Visuospatial	4 (3–5)	4 (3–4)	0.826
Naming	3 (3–3)	3 (3–3)	0.083
Attention	6 (5–6)	6 (5–6)	0.976
Language	2 (1–3)	2 (1–3)	0.617
Abstraction	1 (0–2)	1 (1–2)	0.116
Memory Index Score	13 (9–14)	13 (11–15)	**0.002**
Orientation	6 (6–6)	6 (6–6)	0.083
Score	24 (22–26)	25 (23–27)	**0.02**
Normal cognitive performance (>25)	20 (33.3)	28 (46.7)	
Mild impairment (18–25)	37 (61.7)	30 (50)	
Moderate impairment (10–17)	3 (5)	2 (3.3)	
Severe impairment (<10)	0 (0)	0 (0)	

Values are presented as n (%) for categorical variables, mean ± standard deviation for normally distributed and median (interquartile range 25th–75th) for non-normally distributed continuous variables. Values in bold denote statistical significance with *p* < 0.05.

**Table 4 viruses-17-00135-t004:** Comparison between participants with previous COVID-19 diagnosis and those without.

	Non-COVID-19 (n = 23)	COVID-19 (n = 37)	*p*
Male gender	22 (95.7)	33 (89.2)	0.640
Education < 13 years	9 (39.1)	10 (27)	0.397
HIV acquisition			0.631
MSM	19 (82.6)	29 (78.4)	
PWID	2 (8.7)	1 (2.7)	
Heterosexual	2 (8.7)	5 (13.5)	
Uknown	0 (0)	1 (2.7)	
Other	0 (0)	1 (2.7)	
Psychiatric history	3 (13)	8 (22.2)	0.502
Neurological history	1 (4.3)	1 (2.7)	1
CD4 count at HIV diagnosis	182 (83–519)	386 (270–554)	0.106
Log HIV RNA at HIV diagnosis	5 ± 0.8	4.7 ± 0.9	0.276
Log HIV DNA at HIV diagnosis	2.5 ± 0.8	2.8 ± 0.4	0.416
AIDS diagnosis concurrent with HIV	5/8 (62.5)	4/9 (44.4)	0.637
HIV Treatment			
Backbone			0.911
ABC/3TC	2 (8.7)	4 (10.8)	
TXF/XTC	21 (91.3)	31 (83.8)	
Other	0 (0)	2 (5.4)	
3rd agent			0.533
INSTI	13 (56.5)	23 (62.2)	
Boosted PI	2 (8.7)	3 (8.1)	
NNRTI	6 (26.1)	7 (18.9)	
Other/combination	0 (0)	1 (2.7)	
Depression, median	3 (1–7)	4 (1–5)	0.860
Depression, strata			0.121
0–7	18 (78.3)	35 (94.6)	
8–10	4 (17.4)	1 (2.7)	
11–15	1 (4.3)	1 (2.7)	
≥16	0 (0)	0 (0)	
Anxiety, median	4 (2–8)	4 (2–6)	0.982
Anxiety, strata			0.099
0–7	17 (73.9)	32 (86.5)	
8–10	1 (4.3)	4 (10.8)	
11–15	4 (17.4)	1 (2.7)	
≥16	1 (4.3)	0 (0)	
1st MOCA test			
Age at testing	49.6 ± 10.8	45 ± 9.5	0.09
CD4 at testing	643 (456–928)	861 (612–1067)	0.093
AIDS diagnosis at testing	8 (34.8)	9 (25)	0.557
Years with HIV	7 (4–12)	8 (6–12)	0.374
Years with AIDS	4 (2–5)	8 (2–10)	0.370
Years under treatment	6 (4–11)	7 (5–10)	0.543
Visuospatial	4 (3–5)	4 (3–5)	0.862
Naming	3 (3–3)	3 (3–3)	0.248
Attention	6 (5–6)	6 (5–6)	0.611
Language	2 (1–3)	2 (1–3)	0.911
Abstraction	1 (0–2)	1 (0–2)	0.961
Memory Index Score	13 (9–14)	12 (9–14)	0.326
Orientation	6 (6–6)	6 (6–6)	0.740
MOCA Score	25 (23–26)	24 (22–26)	0.412
Categories			0.971
Normal cognitive performance (>25)	8 (34.8)	12 (32.4)	
Mild impairment (18–25)	14 (60.9)	23 (62.2)	
Moderate impairment (10–17)	1 (4.3)	2 (5.4)	
Severe impairment (<10)	0 (0)	0 (0)	
2nd MOCA test			
Age at testing	52.8 ± 10.8	48.1 ± 9.6	0.086
CD4 at testing	689 (495–899)	817 (665–1035)	0.108
AIDS diagnosis at testing	8 (34.8)	9 (25)	0.557
Years with HIV	10 (8–15)	11 (10–15)	0.213
Years with AIDS	8 (5–9)	11 (5–14)	0.332
Years under treatment	9 (8–14)	10 (7–14)	0.569
Visuospatial	4 (3–4)	4 (3–4)	0.401
Naming	3 (3–3)	3 (3–3)	0.634
Attention	6 (5–6)	6 (5–6)	0.440
Language	2 (1–3)	2 (1–3)	0.444
Abstraction	1 (0–2)	1 (1–2)	0.342
Memory Index Score	13 (10–14)	14 (11–15)	0.304
Orientation	6 (6–6)	6 (6–6)	0.165
MOCA Score	25 (22–27)	24 (23–27)	0.597
Categories			0.891
Normal cognitive performance (>25)	10 (43.5)	18 (48.6)	
Mild impairment (18–25)	12 (52.2)	18 (48.6)	
Moderate impairment (10–17)	1 (4.3)	1 (2.7)	
Severe impairment (<10)	0 (0)	0 (0)	

Values are presented as n (%) for categorical variables, mean ± standard deviation for normally distributed and median [interquartile range 25th–75th] for non-normally distributed continuous variables. *p* values < 0.05 denote statistical significance. COVID-19: coronavirus disease 2019, MSM: men who have sex with men, PWID: people who inject drugs, ABC: abacavir, 3TC: lamivudine, TXF: tenofovir alafenamide or disoproxil, FTC: emtricitabine, INSTI: integrase strand transfer inhibitor, PI: protease inhibitor, NNRTI: non-nucleoside reverse transcriptase inhibitor, MOCA: Montreal Cognitive Assessment.

## Data Availability

The data are available at the Pergamos repository of the University of Athens https://pergamos.lib.uoa.gr/uoa/dl/object/3447186.

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
