# Peer review of "Evaluation of Cognitive Functions in People Living with HIV Before and After COVID-19 Infection"

_viruses, 2025, doi:10.3390/v17010135_

Round 1
Reviewer 1 Report
Comments and Suggestions for Authors
The results from your cohort are interesting, even if it is a small study population. I appreciate the challenges of studying a patient population during the pandemic. Overall, the study is informative and well-written; as such, I don't have major critiques.
A couple of clarifications will help refine the manuscript.
1) How did you account for potential asymptomatic individuals infected with Sars CoV-2 that didn't happen to have a positive test result? What is your assessment of the impact of the asymptomatic individuals in this cohort?
2) These kinds of cognitive assessments are done on an annual basis. I am assuming this cohort has been studied for several years before the pandemic. Why have you chosen only two-time points for MOCA scores? What were the MOCA score trends in the pre-pandemic era? Was the patient population not cogntively examined during the pandemic?
Author Response
Thank you for the encouraging comments and the time spent reviewing our manuscript.
Comment 1: How did you account for potential asymptomatic individuals infected with Sars CoV-2 that didn't happen to have a positive test result? What is your assessment of the impact of the asymptomatic individuals in this cohort?
Response 1: Thank you for this question. Patients were periodically tested for SARS-CoV-2, but it is quite possible that an asymptomatic infection at some point might have escaped diagnosis. In theory we could perform serologic testing, but this would not tell us at what point they had been infected in relation to the MOCA testing. Moreover, based on a metanalysis of 1.2 million cases, it is very unlikely that asymptomatic persons would have lasting cognitive symptoms (JAMA. 2022;328:1604–1615. doi: 10.1001/jama.2022.18931 we added this citation). We have added a few lines discussing this nonetheless, to the limitations section of our study,
Comment 2: These kinds of cognitive assessments are done on an annual basis. I am assuming this cohort has been studied for several years before the pandemic. Why have you chosen only two-time points for MOCA scores? What were the MOCA score trends in the pre-pandemic era? Was the patient population not cogntively examined during the pandemic?
Response 2: We began systematically testing our population in 2019 and the initial plan was for a yearly follow-up. However, the pandemic changed our plans, In Greece, as with other countries, there was an extensive lock down for several months and this made both recruitment for the study and follow-up difficult. Therefore, we only have two timepoints to compare,
Reviewer 2 Report
Comments and Suggestions for Authors
The manuscript titled “Evaluation of cognitive functions in people living with HIV before and after COVID-19 infection” seeks to determine the effect of SARS-CoV-2 infection on cognition in people living with HIV (PLWH). The authors used the Montreal Cognitive Assessment (MOCA) scale to assess cognition and study participants from a Laiko cohort that had been followed for a minimum of 2 years post ART initiation and have sustained virological control, as previously described. The primary outcome, focused on 116 virally suppressed PLWH, was defined as changes in the MOCA scores in relation to HIV history, the secondary outcome, focused on 60 PLWH with two MOCA assessments before and after the pandemic, was defined as changes to MOCA scores overtime +/- SARS-CoV-2 infection. Within the primary outcome, the authors report that there was a significant inverse relation between age and MOCA score and between final score and CD4 counts at the time of testing, with no other significant associations observed. Within the secondary outcome, the authors report CD4 counts were associated with MOCA score in the follow up timepoint only regardless of COVID status and the MOCA scores were correlated between timepoints. There was a slight yet not significant improvement in the MOCA scores between timepoints, regardless of COVID status. Overall, the authors report that SARS-CoV-2 infection did not appear to have an effect and that further work is needed to determine the role of COVID-19 in this population. While this is an interesting paper and important to be reported even with the non-significant outcomes as the longitudinal aspect of this study is significant, there is additional data needed to make this manuscript more complete and alleviate the concerns of this reviewer.
Major:
1. When referring to infection use the term SARS-CoV-2, when referring to the illness use COVID-19, as COVID-19 is not an infection.
2. The abstract reads as those SARS-CoV-2 infection may be beneficial. This is misleading as the finding is not significant and should be removed or edited to report no effect of SARS-CoV-2 infection.
3. We know from previous work that cognition in PLWH starts to improve about 6 months post ART initiation. Though the authors report that the participants were on ART and had been suppressed a minimum of two consecutive timepoints, a timeline is not reported. How long had the participants been on ART at the first and second MOCA assessments?
4. What controls were used to define impairment? Are these well matched to this population?
5. The methods mention measuring HIV DNA in plasma, but the cited paper says PBMCs. Please clarify.
6. COVID pos were defined as having a reported positive PCR test, but it is possible the participants without a test were unaware of being infected or did not report it. This should be listed as a limitation of the study.
7. In the subset of 60 participants with confirmed SARS-CoV-2 infection (n=37), it is unclear how soon after infection cognition was assessed or if there was a percentage of participants that still exhibited symptoms (ie long COVID).
8. The discussion suggests that viral suppression mitigates all damage from infection. While CD4 counts becoming significant at second time point argues that cognition is improving with immune recovery from HIV infection and because of treatment, as expected with suppression, there is no data from before HIV infection so the statement cannot be proven. It needs to be rewritten to fit the data accordingly.
9. Is there any vaccination information on these participants?
Minor
1. Spelling mistakes (line 32 withing, line 59 metanalysis)
Author Response
Thank you for the encouraging comments and the time spent reviewing our manuscript.
Major:
Comment 1: When referring to infection use the term SARS-CoV-2, when referring to the illness use COVID-19, as COVID-19 is not an infection.
Response 1: Thank you for the correction. We have reviewed the manuscript and made appropriate changes.
Comment 2: The abstract reads as those SARS-CoV-2 infection may be beneficial. This is misleading as the finding is not significant and should be removed or edited to report no effect of SARS-CoV-2 infection.
Response 2: We removed the sentence.
Comment 3: We know from previous work that cognition in PLWH starts to improve about 6 months post ART initiation. Though the authors report that the participants were on ART and had been suppressed a minimum of two consecutive timepoints, a timeline is not reported. How long had the participants been on ART at the first and second MOCA assessments?
Response 3; We appreciate the suggestion! This information is available but didn’t occur to us to add it into the tables. We have calculated the relevant median values and Mann Whitney p values and added this information to tables 3 and 4.
Comment 4: What controls were used to define impairment? Are these well matched to this population?
Response 4: Definitions with regards to the cut-off values for MOCA were based on the original work of Dr Nasreddine (citation 15) and on two Greek validation cohorts (citations 19 and 20). Validation in an HIV cohort has been published. We have added a citation (22) in our methods section. We did not have a control population in our study.
Comment 5: The methods mention measuring HIV DNA in plasma, but the cited paper says PBMCs. Please clarify.
Response 5: DNA was measured in PBMCs as described in the cited paper. Mention of DNA measurement in plasma was a mistake, and it was corrected.
Comment 6: COVID pos were defined as having a reported positive PCR test, but it is possible the participants without a test were unaware of being infected or did not report it. This should be listed as a limitation of the study.
Response 6: We agree and this was pointed out by the other reviewer as well. It was added to the limitations of our study as suggested.
Comment 7: In the subset of 60 participants with confirmed SARS-CoV-2 infection (n=37), it is unclear how soon after infection cognition was assessed or if there was a percentage of participants that still exhibited symptoms (ie long COVID).
Response 7: None of the participants had evidence of Long COVID, apart from a potential neurocognitive decline that was under investigation. We do have information on the time frame from COVID to 2nd testing and we added the relevant information in the results section,
Comment 8: The discussion suggests that viral suppression mitigates all damage from infection. While CD4 counts becoming significant at second time point argues that cognition is improving with immune recovery from HIV infection and because of treatment, as expected with suppression, there is no data from before HIV infection so the statement cannot be proven. It needs to be rewritten to fit the data accordingly.
Response 8: This statement is more based on the given citations rather than our own results. However, we understand the possible alluded misunderstanding and we have removed the last sentence in that paragraph in the Discussion section.
Comment 9: Is there any vaccination information on these participants?
Response 9: Vaccination data is not readily available, but it is an interesting question, and we will look into collecting this data for
Comment 10: Spelling mistakes (line 32 withing, line 59 metanalysis)
Response 10: We appreciate your catching these mistakes and we corrected those. We proofread the whole manuscript one more time for anything else we might have missed.